analytical chemistry/organic chemistry/medicinal chemistry

daidzein napsylates, lipophilicity, HAVSMCs, cell absorption and metabolism, structure–property relationship, metabolic pathway *in vitro*

**Author for correspondence:**
You Peng
e-mail: trihydracid@126.com

[†]These authors contributed equally to this paper.

This article has been edited by the Royal Society of Chemistry, including the commissioning, peer review process and editorial aspects up to the point of acceptance.

# Study on pharmacological properties and cell absorption metabolism of novel daidzein napsylates

Yanxiao Jiao[1,†], Jing Peng[5,†], Xinglin Ye[2], Huanan Hu[2], Lijun Gan[3], Jianyuan Yang[4] and You Peng[1]

[1]Jangxi Province Engineering Research Center of Ecological Chemical Industry, [2]College of Chemistry and Environmental Engineering, [3]School of Nursing, and [4]College of Pharmaceutical and Life Sciences, Jiujiang University, Jiujiang, People's Republic of China
[5]College of Traditional Chinese Medicine, Inner Mongolia Medical University, Hohhot, People's Republic of China

YJ, 0000-0003-2678-4723; YP, 0000-0003-0468-4801

Novel daidzein napsylates (DD4 and DD5) were synthesized by microwave irradiation, according to structural modification of daidzein (DAI) using the principle of pharmacokinetic transformation. The pharmacological properties of DD4 and DD5 were evaluated via high performance liquid chromatography (HPLC) and calculated based on the drug design software CHEMAXON 16.1.18. The cell uptake changes of DD4 and DD5 were investigated to analyse the structure–property relationship. The metabolisms of DD4 and DD5 were analysed by HPLC-mass spectrometry in human aortic vascular smooth muscle cells (HAVSMCs) and their possible metabolic pathways were inferred *in vivo*. The results showed that the solubility of DD4 and DD5 was increased by $2.79 \times 10^5$ and $2.16 \times 10^5$ times compared to that of DAI, separately, in ethyl acetate. The maximum absorption rates of DD4 and DD5 were enhanced by 4.3–4.5 times relative to DAI. Preliminary studies on metabolites of DD4 and DD5 in HAVSMCs showed that DD4 and DD5 were hydrolysed into DAI under the action of intracellular hydrolase in two ways, ester hydrolysis or ether hydrolysis. Then, DAI was combined with glucuronic acid to form daidzein monoglucuronate under the action of uridine diphosphate (UDP)-glucuronidase. Meanwhile, it was also found that metabolite M5 of DD5 could undergo glucuronidation under the action of UDP-glucuronosyltransferase and competitive sulphation under the action of sulphotransferase to produce its sulfate conjugate M7. Analysis of structure–property relationships indicated that the absorption and utilization of drugs is closely relative to the physical properties and could be improved by adjusting the

liposolubility. The pharmaceutical properties were optimized comprehensively after DAI was modified by naphthalene sulphonate esterification. This indicates that this kind of derivatives may have relatively good absorption and transport characteristics and biological activities *in vivo*. The research on biological activities of the new derivatives (DD4 and DD5) is ongoing in our laboratory.

# 1. Introduction

Daidzein (DAI, 4′, 7-dihydroxyisoflavone) is a secondary metabolite in soya beans that (i) plays an important role in (a) the treatment of a variety of cancers [1] and (b) the prevention of cardiovascular disease [2], (ii) has hypoglycaemic activity [3], (iii) exerts anti-oxidation and anti-inflammatory effects [4] and positive effects on cerebral ischaemic injury [5], and (iv) has other important physiological functions. However, the lipophilicity of DAI is weak because of two polar hydroxyl groups in its structure. In the meantime, its hydrophilicity is poor, resulting in high lattice energy through intermolecular hydrogen bonds because of hydroxyls. Ysuda *et al.* [6] reported that orally administered DAI was absorbed from the small intestine and metabolized in combination with glucuronide and sulphuric acid under the action of uridine diphosphate (UDP)-glucuronidetransferase and sulphotransferase in the liver. The products, glucuronide and sulphuric acid conjugate, were excreted through urine and bile, and underwent strong hepatointestinal circulation. This suggested that the combined metabolism was its main metabolic pathway. Xu *et al.* [7] found that daidzein-7-glucuronic acid conjugate, daidzein-7-sulphuric acid conjugate and daidzein-7-glucuronic acid-4′-sulphuric acid conjugate were the major forms after multiple intragastric administration of DAI in the plasma of rats. A study by Kulling *et al.* [8] showed that DAI was oxidatively metabolized to monohydroxylated and dihydroxylated products of DAI in the human liver. Qiu's research [9] proved that it was rapidly absorbed and converted into daidzein-7-*O*-glucuronide by the first-pass metabolism after intragastric administration of DAI solution in rats and the absolute bioavailability calculated on the basis of free DAI was 12.8%. Heinonen *et al.* [10] reported the metabolism of DAI in the human body; the metabolites of DAI in urine were analysed as trimethylsilyl derivatives by gas chromatography-mass spectrometry, and the metabolic pathway of DAI in a human was put forward after soya supplementation. In summary, DAI is rapidly absorbed into the body and metabolized and thereby deactivated, and the bioavailability is not ideal, which makes its widespread clinical use difficult. In our group, daidzein napsylates were synthesized to improve their pharmacological properties and bioactivity, using DAI as the lead compound, on the basis of the principle of pharmacokinetic transformation [11]. It is hoped that their pharmaceutical properties can be improved through changes in the structure in order to enhance their biological activity.

Owing to the difficulty of introducing alkyl groups in DAI by traditional heating methods, 7-methoxy-daidzein (DD2) and 7-ethoxy-daidzein (DD3) were synthesized under microwave irradiation to improve the yield of target derivatives, and then DD4 and DD5 were synthesized by esterification with naphthalene sulphonic acid. We investigated the pharmaceutical properties of the synthesized target products DD4 and DD5. Their solubility and apparent lipid-water partition coefficients (lg$P$) were determined by high performance liquid chromatography (HPLC). The properties of DD4 and DD5, such as lg$P$, molecular polar surface area, molecular polarizability, the molecular molar refractive index and the hydrogen bond and dissociation constant (pKa), were calculated by the drug design software CHEMAXON 16.1.18. The absorption and utilization of DD4 and DD5 in human aortic vascular smooth muscle cells (HAVSMCs) were investigated by HPLC. The metabolites of DD4 and DD5 were detected by HPLC-mass spectrometry (HPLC-MS) in HAVSMCs and their possible metabolic pathways were inferred *in vitro* preliminarily. The structure–property relationships of DAI and its derivatives were explored preliminarily according to these results. The synthetic route is shown in scheme 1.

# 2. Experimental procedures

## 2.1. Equipment and reagents

A WRR Digital Display Micro Melting Point Instrument (Beijing Tektronix Instrument Co., Ltd.) (without corrected thermometer), a THZ-22 desktop constant temperature oscillator (Jiangsu Taicang Experimental Instrument Equipment Factory) and an XH-MC-1 type Xianghu laboratory microwave synthesis reactor were used. 1H-nuclear magnetic resonance (NMR) spectra were recorded in

$R$ = 2-naphthalenesulphonyl, $R^1$ = CH$_3$, $R^2$ = CH$_2$CH$_3$

**Scheme 1.** Synthesis of daidzein napsylates.

$d_6$-dimethyl sulphoxide (DMSO) on a Bruker AVANCE III HD 400 MHz spectrometer, using tetramethylsilane as internal standard. Fourier transform-infrared (FT-IR) spectra were recorded on a Bruker VERTEX 70 Fourier transform-infrared FT-IR spectrophotometer. Elemental analysis was recorded on an Elementar Vario Micro Select element analyser. Drug calculation software CHEMAXON 16.1.18., an Agilent 6430 Triple Quadrupole liquid chromatography-MS, an Agilent 1260 HPLC (Agilent 1260 autosampler, G7129A), an EC-C18 column (150 mm × 4.6 mm, 5 µm), a TG18-WS Xiangli tabletop high-speed centrifuge, a Haier ultra-low temperature refrigerator, a DN-24A nitrogen blower, an ESCO carbon dioxide incubator and an Olympus inverted biological phase-contrast microscope were used.

DAI (purity > 98%) was purchased from Shanxi Huike Plant Development Co., Ltd. The mixture of penicillin and streptomycin was obtained from Solarbio. Dimethyl sulphoxide (DMSO, biological reagent) was purchased from Sigma-Aldrich (St. Louis, MO, USA). Foetal bovine serum (South American origin, No.: 11G327) was obtained from Excell. Dulbecco's modified Eagle medium (DMEM) with high glucose (no.: AD16191265) was purchased from Hyclone. HAVSMCs were obtained from Beina Chuanglian Biotechnology Co., Ltd. All other chemicals and reagents were analytical reagent or chemical pure.

## 2.2. Microwave synthesis of daidzein napsylates

### 2.2.1. Microwave synthesis of DD2

DAI (0.9896 g) was dissolved in acetone (55 ml) in a 100 ml round-bottom flask, and the mixture was heated to 60°C. Then potassium carbonate (1.2601 g) was added to the solution and stirred until completely dissolved. Dimethyl sulphate (0.368 ml) was added with vigorous stirring (microwave power of 300 W) and refluxing for 55 min. The reaction was monitored by thin-layer chromatography (TLC) until the starting materials were completely reacted. The crude product was obtained by filtration. The crude product was purified via column chromatography, and the mobile phase was dichloromethane/acetone ($V$(CH$_2$Cl$_2$):$V$(CH$_3$COCH$_3$) = 20 : 1). A white solid was obtained (0.7252 g, yield 69.4%), at m.p. 218–219°C, with an MS $m/z$ value of 269.31 ([M + H]$^+$, 100%), in agreement with literature data [12].

### 2.2.2. Synthesis of DD4

DD2 (0.1036 g) and 2-naphthalenesulphonyl chloride (0.1658 g) were dissolved in pyridine (20 ml) in a 100 ml round-bottom flask at 65°C for 48 h under constant stirring. The mixture was filtered and the filtrate is evaporated under reduced pressure to remove the solvent. The crude product was purified via column chromatography, and the mobile phase was dichloromethane/acetone ($V$(CH$_2$Cl$_2$): $V$(CH$_3$COCH$_3$) = 10:1). A yellow solid was obtained (0.0879 g, yield 49.6%), at m.p. 144–156°C, with an MS $m/z$ value of 459.44 ([M + H]$^+$, 100%); $^1$H NMR (400 MHz, CDCl$_3$) $\delta$ 8.43 (s, 1H, C1''-H), 8.18 (d, $J$ = 8.8 Hz, 1H, C5-H), 8.03 (d, $J$ = 8.8 Hz, 1H, C4''-H), 7.97–7.95 (m, 3H, C2-H, C2'-H, C6'-H), 7.87 (d, $J$ = 8.7 Hz, 1H, C3''-H), 7.73 (t, $J$ = 7.3 Hz, 1H, C7''-H), 7.66 (t, $J$ = 7.3 Hz, 1H, C6''-H), 7.46 (d, $J$ = 7.9 Hz, 2H, C5''-H, C8''-H), 7.30 (s, 1H, C8-H), 7.00–6.95 (m, 3H, C6-H, C3'-H, C5'-H), 3.84 (s, 3H, 7-OCH$_3$); IR (KBr): 3072, 2930, 1632, 1606, 1595 cm$^{-1}$; Anal. Calcd. for C$_{26}$H$_{18}$O$_6$S: C 68.11, H 3.96, S 6.99; found C 67.92, H 4.01, S 6.77.

### 2.2.3. Microwave synthesis of DD3

DAI (0.5476 g) was dissolved in acetone (35 ml) in a 100 ml round-bottom flask, and the mixture was heated to 60°C. A solution of potassium hydroxide (0.2167 g) in water (0.8 ml) was added to the mixture. A solution of dimethyl sulphate (0.368 ml) in acetone (5 ml) was added dropwise with vigorous stirring for 10 min (microwave power of 200 W) and refluxing for 90 min. The reaction was monitored by TLC until the starting materials were completely reacted. The reaction solution was cooled in ice water, extracted with ethyl acetate (25 ml × 3) and dried with anhydrous sodium sulphate overnight. The solvent was removed under reduced pressure. The crude product was purified via column chromatography, and the mobile phase was dichloromethane/acetone ($V(CH_2Cl_2)$: $V(CH_3COCH_3) = 20:1$). A white solid was obtained (0.4389 g, yield 72.3%), at m.p. 131–133°C, with an MS $m/z$ value of 283.46 ($[M + H]^+$, 100%), in agreement with literature data [13].

### 2.2.4. Synthesis of DD5

DD3 (0.1463 g) and 2-naphthalenesulphonyl chloride (0.1593 g) were dissolved in pyridine (15.6 ml) in a 100 ml round-bottom flask at room temperature for 48 h under constant stirring. The mixture was filtered and the filtrate was evaporated under reduced pressure to remove the solvent. The crude product was purified via column chromatography, and the mobile phase was dichloromethane/acetone ($V(CH_2Cl_2):V(CH_3COCH_3) = 10:1$). A yellow solid was obtained (0.0359 g, yield 14.9%), at m.p. 151–153°C, with an MS $m/z$ value of 473.48 ($[M + H]^+$, 100%);1H NMR (400 MHz, CDCl$_3$) $\delta$: 8.43 (s, 1H, C1″-H), 8.18 (d, $J = 8.6$ Hz, 1H, C5-H), 8.03 (d, $J = 8.5$ Hz, 1H, C4″-H), 7.96 (d, $J = 8.5$ Hz, 2H, C2′-H, C6′-H), 7.94 (s, 1H, C2-H), 7.87 (d, $J = 8.7$ Hz, 1H, C3″-H), 7.73 (t, $J = 7.4$ Hz, 1H, C7″-H), 7.66 (t, $J = 7.6$ Hz, 1H, C6″-H), 7.44 (d, $J = 8.1$ Hz, 2H, C5″-H, C8″-H), 7.29 (s, 1H, C8-H), 6.99–6.94 (m, 3H, C6-H, C3′-H, C5′-H), 4.06 (q, $J = 7.0$ Hz, 1H, 7-OCH2-), 1.43 (t, $J = 6.9$ Hz, 1H, -CH3); IR (KBr): 2981, 2926, 1636, 1625, 1607 cm$^{-1}$; Anal. Calcd. for $C_{27}H_{20}O_6S$: C 68.63, H 4.27, S 6.79; found C 68.44, H 4.29, S 6.51.

# 3. Pharmacological properties

The solubilities and lg$P$ of DD4 and DD5 were determined by HPLC, and their molecular polar surface area and polarizability, etc. were calculated by the software CHEMAXON 16.1.18 to predict their transport process and biological activity in the body.

## 3.1. Chromatographic conditions

Mobile phase: acetonitrile-methanol (1 : 1); flow rate: 1.0 ml min$^{-1}$; injection volume: 20 μl; wavelength: 230 nm or 254 nm; column temperature: 25°C.

## 3.2. Preparation of standard solution

DD4 was dissolved in methanol to prepare a DD4 methanol solution with a concentration of 204 μg ml$^{-1}$, which was stored at 4°C as the stock solution. DD4 stock solution was diluted with methanol to prepare a series of standard DD4 solutions with concentration of 0.408, 1.02, 2.04, 4.08, 8.16 and 16.32 μg ml$^{-1}$ and stored at 4°C for future use. A series of standard DD5 solutions of 0.356, 0.890, 1.78, 3.56, 7.12 and 14.24 μg ml$^{-1}$ were prepared and stored by the same method.

## 3.3. Establishment of the detection method

### 3.3.1. Investigation of the linear relationship

A series of standard solutions of DD4 and DD5 (see §3.2) were analysed under chromatographic conditions as described in §3.1. The regression equations of DD4 and DD5 were obtained via regression calculation by weighted least-squares methods, by taking the concentration of the analyte

as the abscissa $X$ and the peak area as the ordinate $Y$. The following formulae were obtained:

$$Y = 174.31X + 116.99, \ r^2 = 0.997,$$

$$Y = 132.23X + 127.41, \ r^2 = 0.996.$$

The linear ranges were 0.408–16.32 µg ml$^{-1}$ and 0.356–14.24 µg ml$^{-1}$, respectively.

### 3.3.2. Lower limit of quantitation and lower limit of detection

A solution with a concentration of 408 ng ml$^{-1}$ of DD4 was prepared following the method in §3.2, which was analysed through five samples. The concentration of each sample was obtained according to a standard curve to determine the intra-day precision (relative standard deviation; RSD) and accuracy (relative error; RE) of each component at this concentration (see the electronic supplementary material, table S1). The results showed that both RSD and RE were within a reasonable range, and the limit of quantitation (LOQ) of DD4 was confirmed to be 408 ng ml$^{-1}$. In our HPLC analysis, the signal-to-noise ratio (S/N) was 10 at this concentration. The LOQ of DD5 was 356 ng ml$^{-1}$, which was determined in the same way. The RSD and RE of DD5 under LOQ concentrations are shown in the electronic supplementary material, table S1. The sample of DD4 with a concentration of 408 ng ml$^{-1}$ was diluted 12 times and analysed. The S/N was 3 by HPLC, which indicated the limit of detection (LOD) of DD4 was 34 ng ml$^{-1}$. The LOD of DD5 was 59 ng ml$^{-1}$ using the same method.

### 3.3.3. Precision and accuracy of the method

Quality control (QC) samples of DD4 and DD5 were prepared with low, medium and high concentrations, and each solution was determined through six samples. It was continuously measured for 3 days to obtain the concentration of QC samples. The RSD and RE of each component were calculated by comparing the measurement results with the concentrations of the preparations. The results are shown in the electronic supplementary material, table S2. The intra-day precision of DD4 and DD5 was within 12%, the inter-day precision was less than 13%, and the accuracy was between −6.7% and 5.9%, which were within rational limits.

## 3.4. Determination of solubility and apparent lipid-water partition coefficient

### 3.4.1. Solubility

Five millilitres of water, methanol, cyclohexane and ethyl acetate was added to test tubes, and then samples (DD4 or DD5) were added. These solutions were shaken at constant temperature (37 ± 0.1)°C for 48 h. The supernatant was taken to determine its solubility with HPLC [14]. Solutions of tested sample beyond the linear concentration range were diluted with methanol for determination.

### 3.4.2. Apparent lipid-hydro partition coefficient

DD4 (5.1 mg) was dissolved in 50 ml of water-saturated $n$-octanol solution to prepare a solution with a concentration of 102 mg l$^{-1}$. Solution (2 ml) and water-saturated $n$-octanol solution (2 ml) were added to a plugged 10 ml tube. The prepared sample solution was placed in a shaker and shaken to equilibrium overnight in the dark at constant temperature (37 ± 0.1)°C. The lower aqueous phase was taken and analysed by HPLC after centrifugation at a speed of 3000 r min$^{-1}$ for 10 min. The drug concentration in the aqueous phase ($\rho_w$) was calculated via the standard curve according to the peak area. DD5 (4.5 mg) was dissolved in 50 ml of water-saturated $n$-octanol solution to prepare a solution with a concentration of 90 mg l$^{-1}$. Other methods were the same as those for DD4 [14].

# 4. Drug absorption in human aortic vascular smooth muscle cells

## 4.1. Cell culture and drug treatment

HAVSMCs were obtained from resuscitation in cryopreservation tubes. HAVSMCs at passages 3–10 were seeded at a density of $10^5$ in 6-well plates, 2 ml per well. The cells were cultured at 37°C in a 5% carbon dioxide incubator for 24 h to adhere to the cell wall; cells were cultured in serum-free DMEM for 24 h.

The cells were synchronized, the supernatant was carefully aspirated and different concentrations of drug solution were added to the DMEM culture medium containing 10% serum. Equal amounts of DMEM culture medium without drugs were added to the blank group. Each group was set up in triplicate and further cultured. The cultures were terminated at 0.5, 1, 6, 12, 24 or 48 h. Cell culture supernatants were collected and stored at −80°C for further analysis.

## 4.2. The pre-treatment of cell samples

Cell supernatants were thawed at room temperature and vortexed for 1 min. To 50 µl of the cell supernatant, 50 µl of methanol was added and the mixtures were mixed uniformly by eddy current. Ethyl acetate (1 ml) was added to the mixture and vortexed for 3 min. Ethyl acetate was separated by cooled centrifugation at 3000 r min$^{-1}$ and dried with nitrogen, and the residue was dissolved in 500 µl of methanol before injection for analysis.

## 4.3. Chromatographic conditions

Mobile phase: methanol-phosphoric acid solution (0.4%) (1 : 1); flow rate: 0.5 ml min$^{-1}$; injection volume: 10 µl; wavelength: 230 nm or 254 nm; column temperature 25°C.

## 4.4. Preparation of series standard solutions

DD4 was dissolved in methanol to prepare a DD4 methanol solution with a concentration of 4.1 mg ml$^{-1}$, which was stored at 4°C as a stock solution. DD4 stock solution was diluted with methanol to prepare a series of standard DD4 solutions with concentrations of 0.2975, 0.595, 1.19, 5.95 and 11.9 µg ml$^{-1}$ and stored at 4°C for future use. A series of standard DD5 solutions of 0.2675, 0.535, 1.07, 5.35 and 10.7 µg ml$^{-1}$ were prepared and stored by the same method.

## 4.5. Establishment of liquid chromatography for the determination of drug concentration in biological samples [15]

### 4.5.1. Investigation of the linear relationship

DD4 or DD5 standard solutions (25 µl) with different concentrations (see the preparation of series standard solutions in §4.4) were added to 25 µl of cell supernatant and mixed by eddy current. The other procedures were the same as the pre-treatment of cell samples described in §4.2. The regression equations of DD4 and DD5 were obtained via regression calculation by weighted least-squares methods, by taking the concentration of the analyte as the abscissa $X$ and the peak area as the ordinate $Y$. The following formulae were obtained:

$$Y = 47.205X + 11.897, r^2 = 0.9964,$$
$$Y = 63.949X - 1.8763, r^2 = 0.9992.$$

The linear ranges were 0.2975–11.9 µg ml$^{-1}$ and 0.2675–10.7 µg ml$^{-1}$, respectively.

### 4.5.2. Lower limit of quantitation and lower limit of detection

Five solutions with a concentration of 297.5 ng ml$^{-1}$ containing DD4 were prepared, and the concentration of each sample was calculated in accordance with the standard curve to determine the intra-day precision (RSD) and accuracy (RE) of each component at this concentration (see the electronic supplementary material, table S3). The results showed that both RSD and RE were within a reasonable range, and the LOQ of DD4 was confirmed to be 297.5 ng ml$^{-1}$. In our HPLC analysis, the S/N was 10 at this concentration. The LOQ of DD5 was 267.5 ng ml$^{-1}$, which was determined in the same way. The RSD and RE of DD5 under LOQ concentrations are shown in the electronic supplementary material, table S3. The DD4 sample with a concentration of 297.5 ng ml$^{-1}$ was diluted two times and determined. The S/N was 3 by HPLC, which indicated the LOD of DD4 was 148.8 ng ml$^{-1}$. The LOD of DD4 was 17.8 ng ml$^{-1}$ using the same method.

### 4.5.3. Precision and accuracy of the method

Standard solutions of DD4 or DD5 (25 µl) were added to 25 µl of cell supernatant. Samples of DD4 and DD5 were prepared with low, medium and high concentrations, and each solution was determined through six samples. It was continuously measured for 3 days to obtain the concentration of QC samples. The RSD and RE of each component were calculated by comparing the measurement results method with the concentrations of the preparations. The results are shown in the electronic supplementary material, table S4. The intra-day precision of DD4 and DD5 was within 13%, the inter-day precision was less than 14%, and the accuracy was between −12.0% and 10.5%, which were within rational limits.

## 4.6. Pre-treatment and concentration determination of cell samples

Cell supernatants were thawed and vortexed for 1 min. Drug-added supernatant (50 µl) was treated by the same method as the pre-treatment of cell samples described in §4.2.

## 4.7. Data processing

The drug concentrations in the cell supernatants were calculated according to the standard curve established for each analytical batch. The experimental data were statistically analysed by the Student $t$-test. $p < 0.05$ was considered to indicate a significant difference; $p < 0.01$ was considered to indicate an extremely significant difference; $p > 0.05$ indicates no significant difference.

# 5. Cell metabolism

## 5.1. Cell culture and drug treatment

See §4.1.

## 5.2. Analysis conditions

### 5.2.1. Chromatographic conditions

Mobile phase: methanol-formic acid (1 : 1); flow rate: 0.5 ml min$^{-1}$; injection volume: 10 µl; wavelength: 254 nm; column temperature: 25°C.

### 5.2.2. Mass spectrometry conditions

A triple quadrupole tandem precision mass spectrometer (Agilent 6430) with an electron ionization source and a negative ion mode was used for precise determination. The full mass spectrum data were obtained in the range of $m/z$ 50–1000. The optimal mass spectrum parameters were as follows: the capillary voltage was 4.0 kV; the injection cone voltage was 35 V; the desolvation gas flow ($N_2$) rate was 900 l h$^{-1}$; the injection cone gas flow ($N_2$) rate was 50 l h$^{-1}$; the desolvation temperature was 350°C; the ion source temperature was 150°C; the collision gas in collision induced dissociation (CID) mode was helium with high purity; the inlet pressure was set at 40 psi; the CID collision energy of the standard was 5–35 eV and the cracking voltage was 135 V.

## 5.3. Determination of cell supernatant components

The cell supernatant was thawed and vortexed for 0.5 min. Cell supernatant (50 µl) and methane (50 µl) were added to a 2 ml centrifuge tube and vortexed for 3 min. Ammonium dihydrogen phosphate (100 µl, 0.4%) and 1 ml of ethyl acetate were added, and the sample was vortexed for 3 min. The mixture was centrifuged at 3000 r min$^{-1}$ for 10 min. The supernatant (95 µl) was aspirated and dried by $N_2$ flow. Methane (500 µl) was added and the sample was vortexed for 0.5 min. After filtration over a 0.45 µm organic membrane, HPLC-MS analysis was carried out.

## 5.4. Component determination of cell lysate

Cells from the −80°C stock were thawed at room temperature, and 1 ml of thawed cell lysate was added into each well of a 6-well plate. From each group, 0.5 ml of cell lysate was taken; 50 µl of methane, 100 µl of ammonium dihydrogen phosphate (0.4%) and 1 ml of ethyl acetate were added and samples were vortexed for 3 min. Lysate (95 µl) was aspirated and dried by $N_2$ flow. Methane (500 µl) was added and samples were vortexed for 0.5 min. After filtration over a 0.45 µm organic membrane, HPLC-MS analysis was carried out.

## 5.5. Data analysis

Mass spectrometry data acquisition and processing and qualitative analyses were carried out using Mass Hunter acquisition b.03.01.

# 6. Results

## 6.1. Solubility and apparent lipid-hydro partition coefficient

The results of solubility and apparent fat water partition coefficient are shown in table 1. Compared with DAI, the solubility of DD4 and DD5 in all solvents is greatly improved except in water. The solubility of DAI, DD4 and DD5 in water is too poor to be detected. In cyclohexane, the solubility of DD4 was 509.2 µg ml$^{-1}$, and that of DD5 was 69.74 µg ml$^{-1}$, while the solubility of DAI in cyclohexane could not be determined. In ethyl acetate, the solubility of DD4 reached $3.63 \times 10^4$ µg ml$^{-1}$, which was $2.79 \times 10^5$ times higher than that of DAI. The solubility of DD5 in ethyl acetate was also increased by $2.16 \times 10^5$ times relative to that of DAI. It is illustrated that the liposolubility of derivatives is immensely enhanced.

## 6.2. Absorption and utilization of DAI and its derivatives by human aortic vascular smooth muscle cells

The HPLC spectrum of the blank HAVSMC supernatant is shown in figure 1a; the HPLC spectrum of the HAVSMC supernatant treated with DD4 for 0.5 h is shown in figure 1b (retention time 11.418 min); the HPLC spectrum of the HAVSMC supernatant treated with DD5 for 0.5 h is shown in figure 1c (retention time 13.217 min).

The absorption and utilization rates of DD4 and DD5 were calculated by the formula (6.1), with the reduction of drug concentration in the cell culture medium taken as the absorption amount of drug by cells. The initial drug concentration was $c_0$, and the concentration in the cell supernatant at other time points is denoted as $c$. The results are shown in figure 2 and table 2. The absorption of DD4 peaked at 1 h, the drug concentration in the cell supernatant decreased to 20.62 µg ml$^{-1}$, and the absorption rate reached 87.97%. The maximum absorption rate of DD5 at 1 h was 90.34%. With increasing time, the cell absorption rate of the drug was maintained at about 60%. The main reason for the differences in absorption of DD4 and DD5 might be the length of the substituent group (methyl, ethyl) on the 7-position:

$$\text{absorption rate} = \frac{c_0 - c}{c_0} \times 100\%. \tag{6.1}$$

## 6.3. Metabolism of DD4 and DD5 by human aortic vascular smooth muscle cells

### 6.3.1. Cellular metabolic pathway of DD4

The lysates of cells, which were incubated with DD4 for 24 h, were extracted and analysed by HPLC-MS. Four metabolites were found. Their relative molecular masses were obtained using Qualitative Analysis B.03.01 to extract specific molecular chromatograms, which were compared with the mass spectra of blank control samples, as shown in table 3. Four ion peaks, at 429 (1), 267 (2), 253 (3) and 443 (4), were found in the lysate of cells treated with DD4. These four components were also found in the supernatant of cells treated with DD4. The total ion chromatograms and mass spectra of metabolites in lysates of DD4-treated cells are shown in figure 3. The total ion chromatograms and mass spectra of metabolites in supernatants of DD4-treated cells are shown in the electronic supplementary material figure S1.

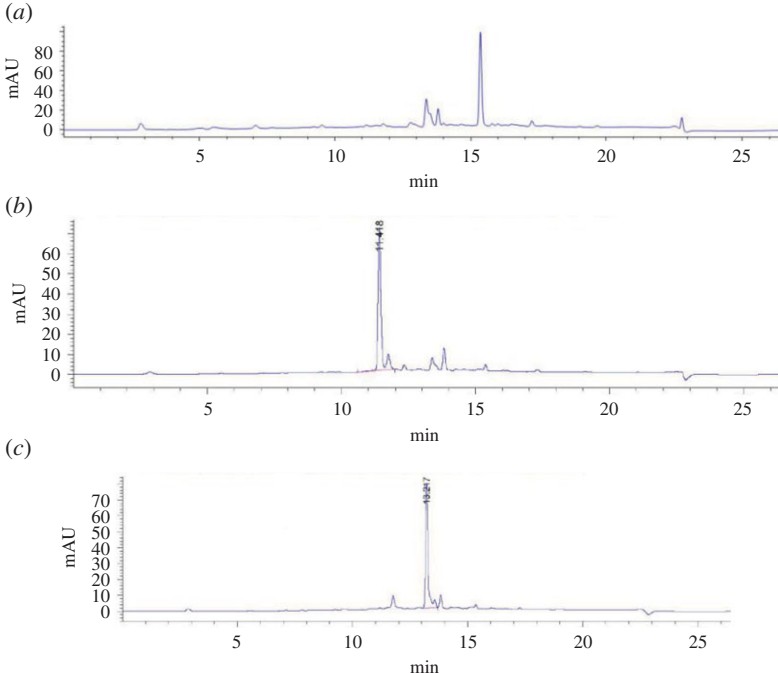

**Figure 1.** HPLC results of cell supernatant of DAI and its derivatives.

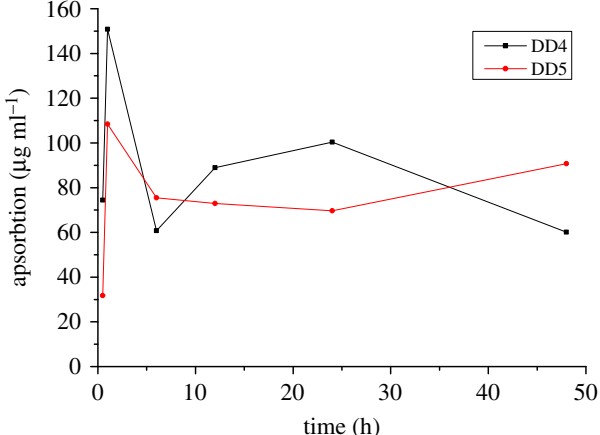

**Figure 2.** The absorption of DD4 and DD5 by HAVSMCs ($n = 3$).

**Table 1.** Solubility and partition coefficients of DAI and its analogues DD4 and DD5.

| compounds | solubility ($\mu g\ ml^{-1}$) | | | | lgP |
| --- | --- | --- | --- | --- | --- |
| | water | methanol | hexane | ethanoate | |
| DAI[a] | —[b] | 97.6 ± 1.24 | —[b] | 0.13 ± 0.11 | —[b] |
| DD4 | —[b] | 824.4 ± 19.3 | 509.2 ± 48.97 | 36 302 ± 4660 | —[b] |
| DD5 | —[b] | 2471 ± 72.4 | 69.74 ± 20.8 | 28 115 ± 1151 | —[b] |

[a]The solubility and the lipo-hydro partition coefficient of DAI can be found in [14].
[b]Cannot be detected.

The possible metabolic pathways of DD4 are preliminarily inferred in scheme 2 according to the cellular metabolic components. DD4 can undergo sulphonate bond disconnection to produce M1 ($m/z$ 267) or hydrolytic demethylation to produce M2 ($m/z$ 443) in cells in light of the mass spectra of cell

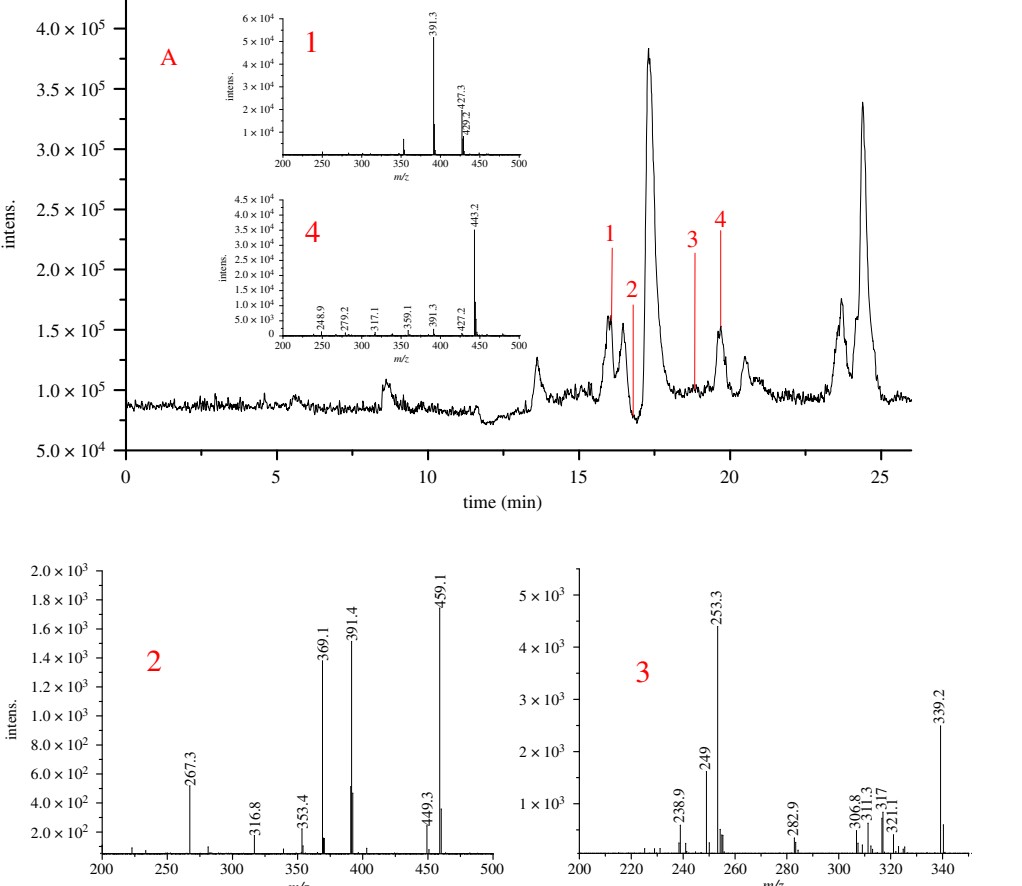

**Figure 3.** The total ion chromatograms and mass spectrum of lysates of cells treated with DD4. **A**: Total ion chromatography; **1**: daidzein-4'-glucoside or daidzein-7-glucoside; **2**: 4'-methyldaidzein; **3**: daidzein (DAI); **4**: daidzein-4'-naphthalene sulphonate.

**Table 2.** The absorption of DAI derivatives by cells.

| | DD4 | | DD5 | |
| --- | --- | --- | --- | --- |
| time (h) | value ($\mu$g ml$^{-1}$) | absorption (%) | value ($\mu$g ml$^{-1}$) | absorption (%) |
| 0.5 | 96.97 ± 15.57 | 43.43 | 88.31 ± 4.25 | 26.43 |
| 1 | 20.62 ± 0.71 | 87.97 | 11.59 ± 0.08 | 90.34 |
| 6 | 110.70 ± 4.98 | 35.43 | 44.60 ± 3.91 | 62.85 |
| 12 | 82.48 ± 3.00 | 51.88 | 47.13 ± 15.99 | 60.74 |
| 24 | 71.01 ± 34.97 | 58.57 | 50.34 ± 1.31 | 58.06 |
| 48 | 111.28 ± 1.72 | 35.09 | 29.29 ± 2.50 | 75.60 |

**Table 3.** Mass spectrum of DD4 metabolites *in vitro*.

| no. | $t_R$ (min) | m/z | formula | chemical name | metabolites |
| --- | --- | --- | --- | --- | --- |
| **1** | 16.077 | 429.2 | **$C_{21}H_{18}O_9$** | daidzein-4'-glucoside or daidzein-7-glucoside | M4 or/and M4a |
| **2** | 16.795 | 267.3 | **$C_{16}H_{12}O_4$** | 4'-methyldaidzein | M1 |
| **3** | 18.831 | 253.3 | **$C_{15}H_{10}O_4$** | daidzein | DAI |
| **4** | 19.689 | 443.2 | **$C_{22}H_{20}O_{10}$** | daidzein-4'-naphthalene sulphonate | M2 |

**Scheme 2.** Possible metabolic pathways of DD4.

metabolism and scheme 2. Although M3, produced from the glucuronidation of M1, has the same molecular weight as M2, it cannot be judged from molecular weight alone whether it is the demethylation hydrolysis product of DD4 or the glucuronidation product of M1. It can be inferred from figure 3 that if M3 is generated by M1 under the action of UDP-glucuronosyltransferase, the retention time of metabolites M2 and M3 is obviously different owing to the great difference in polarity after passing through the reversed-phase column, and the retention time of M3 should be much less than that of M2. While the polarity of M2 was less than that of DAI, its retention time was longer than that of DAI (18.831 min). As seen from table 3, the retention time of the metabolite with $m/z$ 443.1 is 19.689 min, so the metabolite is M2. This means that M1 cannot be transferred to M3 through combined metabolism in cells. DAI could be formed from the further hydrolysis of M1 and M2 and undergoes glucuronidation to produce M4 or/and M4a.

## 6.4. Cellular metabolic pathway of DD5

The lysate of cells which were incubated with DD5 for 24 h was extracted and analysed by HPLC-MS. Six metabolites were found and the relative molecular mass of these metabolites was obtained using Qualitative Analysis B.03.01 to extract specific molecular chromatograms and comparing them with mass spectra of blank control samples, as shown in table 4. Six ion peaks, at 361 (1), 457 (2), 429 (3), 253 (4), 443 (5) and 281 (6), were found in the lysate of cells treated with DD5. These six components were also found in the supernatant of cells treated with DD5. The total ion chromatograms and mass spectra of metabolites in the lysate and supernatant cells treated with DD5 are shown in the electronic supplementary material figures S2 and S3, respectively.

The possible metabolic pathways of DD5 are preliminarily inferred in scheme 3 according to the cellular metabolic components. On the basis of the mass spectra of cell metabolism and scheme 3, M6 ($m/z$ 457) was produced from glucuronidation of M5, which was formed by hydrolysis of

**Scheme 3.** Possible metabolic pathways of DD5.

**Table 4.** Mass spectrum of DD5 metabolites *in vitro*.

| no. | $t_R$ (min) | m/z | formula | chemical name | metabolites |
|---|---|---|---|---|---|
| 1 | 13.573 | 361.3 | $C_{17}H_{14}O_7S$ | daidzein-7-ethyl-4′-sulphonic acid | M7 |
| 2 | 15.781 | 457.3 | $C_{23}H_{22}O_{10}$ | daidzein-7-ethyl-4′-glucoside | M6 |
| 3 | 15.991 | 429.2 | $C_{21}H_{18}O_9$ | daidzein-4′-glucoside or daidzein-7-glucoside | M4 or/and M4a |
| 4 | 18.784 | 253.2 | $C_{15}H_{10}O_4$ | daidzein | DAI |
| 5 | 19.634 | 443.1 | $C_{25}H_{16}O_6S$ | daidzein-4′-naphthalene sulphonate | M2 |
| 6 | 20.937 | 281.3 | $C_{17}H_{14}O_4$ | 7-ethyldaidzein | M5 |

DD5 in cells. It was also found that sulphation of M5 could produce M7 (*m/z* 361), while sulphation in DD4 was not found.

# 7. Discussion

## 7.1. Structure–property relationship

The chemical structural formulae and three-dimensional models of DAI, DD4 and DD5 are shown in figure 4, and the experimental and calculated results of structural characteristics and pharmaceutical properties are shown in table 5.

The original drug DAI was esterified to increase its liposolubility, in order to facilitate the crossing of the cell membrane and hence improve its oral bioavailability and biological activity. An appropriate

**Figure 4.** Structural formula and minimized energy conformation of DAI, DD4 and DD5.

**Table 5.** Structural characteristics and pharmaceutical properties of DAI, D3 and D4.

| characteristics and properties | | DAI[a] | DD4 | DD5 |
|---|---|---|---|---|
| relative molecular mass | | 254 | 458 | 472 |
| melting point (°C) | | 315–323 (decompose) | 144–146 | 151–153 |
| crystal shape | | crystallized powder | crystallized powder | crystallized powder |
| solubility in water (μg ml$^{-1}$) | | ___[b] | ___[b] | ___[b] |
| solubility in ethyl acetate (μg ml$^{-1}$) | | 0.13 ± 0.11 | 36 302 ± 4660 | 28 115 ± 1151 |
| apparent lipid-hydro | calculated value[c] | 2.73 | 5.39 | 5.74 |
| partition coefficient | found | ___[b] | ___[b] | ___[b] |
| molecular polar surface area/ two-dimensional (Å$^2$) | calculated value[c] | 66.76 | 78.90 | 78.90 |
| molecular surface area/ three-dimensional (Å$^2$) | calculated value[c] | 315.07 | 589.83 | 620.77 |
| dissociation constant[d] (%) | calculated value[c] | 10.41 | 100 | 100 |
| polarizability | calculated value[c] | 26.21 | 49.68 | 51.52 |
| refractivity | calculated value[c] | 69.70 | 123.54 | 128.28 |
| H-bond[c] | donor site/count | 2/2 | 0/0 | 0/0 |
| | accepter site/ count | 5/4 | 10/5 | 10/5 |
| absorptivity (%) | | 20.29 | 87.97 | 90.34 |

[a]Structural characteristics of DAI can be found in [16].
[b]Cannot be detected.
[c]At pH 7.4, 410 K.
[d]Percentage of drug prototypes.

relative molecular mass is an important factor in the oral absorption of drugs. It is shown in table 5 that relative molecular weights of DD4 and DD5 are 458 and 472, respectively, which are less than 500. This conforms to the Lipinski rule of a fivefold ratio [16]. The melting points of DD4 and DD5 were significantly lower than those of the original drug DAI, and were in the appropriate range of drug melting points of 100–200°C.

Compared with DAI, the number of hydrogen bond donor sites of DD4 and DD5 decreased from two to zero, but the number of hydrogen bond acceptor sites increased from five to ten (table 5), which illustrated that the hydrogen bonding ability of the derivatives was enhanced. The molar refractive index of DD4 and DD5 increased to between 40 and 130, which conformed to the Lipinski rule of a fivefold ratio [16].

The polarizability of DD4 and DD5 increased in comparison with that of DAI (table 5). It can be seen from figure 4 that a positive charge centre is formed in the molecule after the introduction of a naphthalene sulphonate group into DAI. The deformability of the molecule increases owing to the great polarizability of the S atom, which is conducive to the complementary binding of the drug molecule with the receptor protein in an appropriate configuration to produce the pharmacodynamic effect [17–19]. It is indicated that the introduction of a naphthalene sulphonate group into DAI can optimize its pharmacodynamic conformation, increase the polarizability and deformability of the molecule and hence facilitate the interaction between derivative molecules and receptor proteins to improve the activity.

The calculated results show that the molecular polar surface areas (two-dimensional) of DD4 and DD5 are both 78.9 Å$^2$ (table 5). This is higher than that of DAI, which may reduce the lipid permeability compared with DAI. However, the molecular polar surface areas (two-dimensional) of DD4 and DD5 are within 120 Å$^2$. In the meantime, an acidic hydroxyl group is enclosed in DD4 and DD5 relative to DAI, which reduces the pKa value of the molecule to make it possible to improve the absorption rate on the contrary. As shown in table 2, the maximum absorption rates of DD4 and DD5 increased from 20.29% (DAI) to 87.97% and 90.34%, respectively.

As shown in tables 2 and 5, the cell absorption rates of DD4 and DD5 were greatly increased; they were 4.3 and 4.5 times as high as that of DAI. A naphthalene sulphonate group is introduced at the 7-position of DAI and a methyl or ethyl group is introduced at the 4'-position via a chemical modification to improve the liposolubility, which is conducive to drug absorption. It means that derivatives with high lgP values have high liposolubility and could penetrate cell membranes easily.

In conclusion, the pharmaceutical properties of the naphthalene sulphonate derivatives of DAI are obviously optimized. DD4 and DD5 have moderate relative molecular weights, increased lipophilic hydrophilicity, low pKa values, appropriate molecular polar surface areas, an enhanced hydrogen bonding ability and increased polarizability and deformability. This indicates that the derivatives may exhibit good absorption, transmembrane transport, bioavailability and biological activity.

## 7.2. Optimization of solubility and pharmaceutical properties

Intensive research has been carried out in order to optimize the pharmaceutical properties, improve the bioavailability and enhance the bioactivity of DAI by increasing its solubility. Researchers have performed a series of work to improve the water solubility of DAI. Borghetti *et al.* [20] prepared the daidzein/hydroxypropylβ-cyclodextrin/polyvinylpyrrolidone (DAI/HPβ-CD/PVP) complex to improve the water solubility of DAI; the water solubility of DAI/HPβ-CD/PVP was found to be 12.7 times as high as that of DAI. Jiang *et al.* [21] synthesized 4-oxoacetic acid-7-hydroxyethoxy-daidzein to improve the solubility of DAI, and the solubility reached 54.28 µg ml$^{-1}$ in water with a pH of 7.8. Deng *et al.* [22] prepared solid inclusion complexes of mono-6-amino-yl-cyclodextrin and mono-6-ethylenediamine-diamine cyclodextrin of DAI, whose solubility in water increased about 1800- and 1500-fold at 25°C, respectively. Zhang *et al.* [23] synthesized the derivative sodium daidzein sulphonate via sulphonation, which is easily soluble in water. The anti-hypoxic-ischaemic effect of sodium daidzein sulphonate is obviously better than that of DAI, but the lipid solubility of sodium daidzein sulphonate is not very high, which means its druggability is still not optimistic. The water-soluble inclusion complexes or derivatives were prepared to improve the water solubility of DAI in the above research, without considering the lipid solubility of the derivatives. Only when the drugs have appropriate lipid and water solubilities can the absorption and utilization of drugs be improved. Much work has been done in our group to synthesize a series of benzenesulphonate derivatives of DAI, aiming to improve their lipid solubility. Analysis of their pharmaceutical properties showed that the lipid solubility of derivatives was greatly improved, while the water solubility of some compounds was also significantly increased [14,16,24]. In this study, the solubility of DD4 and DD5, naphthalene sulphonic acid derivatives of DAI, in ethyl acetate was increased by $2.79 \times 10^5$ and $2.16 \times 10^5$ times, respectively, compared with that of DAI. Their appropriate melting point, the improved solubility, the lower dissociation constant, the moderate molecular weight, the appropriate polar surface area of the molecule and the enhanced hydrogen bonding ability predict that this kind of molecule might have a better transmembrane absorption and transport, which was further confirmed by the cell absorption experiments in this study. Therefore, the pharmaceutical properties of DAI can be optimized from two aspects, i.e. by improving the lipid solubility or the water solubility, as the absorptivity of drugs is not only related to the lipid solubility but also to the water solubility of drugs. Better lipid solubility and water solubility can improve the bioavailability of drugs, which needs to be comprehensively analysed in further studies.

## 7.3. Metabolism

*In vitro* metabolic studies can reasonably predict the *in vivo* pharmacokinetic behaviour of candidate compounds by using the *in vitro* metabolic parameters of candidate compounds in the early stage of new drug development [25]. It not only guides the selection of models for later pharmacodynamics, pharmacokinetic and safety evaluations, but also narrows the screening scope of *in vivo* studies, which has broad application prospects [25–29]. The method of using cell lines to study drug metabolism *in vitro* can eliminate the interference of many factors *in vivo* and provide a reliable theoretical basis for the overall experiment. *In vitro* cell cultures are relatively simple in composition, which facilitates the study of inference with metabolic pathways. Metabolites are easy to enrich, isolate and purify, which is conducive to the further study of metabolites. In this article, the preliminary analysis of drug metabolic components in HAVSMCs revealed no redox products of DD4 and DD5, which means that redox enzymes were not active in HAVSMCs. However, under the action of various intracellular hydrolases (such as lipase), hydrolysis of sulphonate and methyl (ethyl) ether occurs in DD4 and DD5 to produce corresponding products, and finally DAI is produced by hydrolysis, which indicates that hydrolase was active in the system. The hydroxyl group in DAI underwent glucuronidation to produce daidzein monoglucuronides M4 and M4a ($m/z$ 429) by the action of UDP-glucuronosyltransferase, and no sulphation products of DAI were found. This showed that the concentration of DAI is higher in the cell system. Because they are competitive reactions, glucuronic acid-binding reactions generally take place at higher substrate concentrations [30]. However, not both of the hydroxyl groups are glycosylated; a possible reason for this is the steric hindrance effect. According to scheme 3, the 4′-hydroxyl group of the metabolite M5 can also undergo glucuronidation to produce M6 ($m/z$ 457) by the action of UDP-glucuronosyltransferase. At the same time, it was found that the metabolite M7 ($m/z$ 361) was produced from M5 via sulphation by the action of sulphotransferase, indicating that the sulphotransferase in the system still has certain activity. This means that when the substrate concentration in the system is moderate, the glucuronide binding reaction and the sulphation reaction can occur simultaneously.

## 8. Conclusion

In this study, two novel naphthalene sulphonate derivatives of DAI were successfully synthesized by microwave irradiation, which shortened the reaction time and increased yield. In the study of the pharmacological properties of the derivatives, compared to DAI, the solubility and cell absorption was increased greatly in HAVSMCs owing to the lipophilic modification of DAI. The pharmaceutical properties were optimized comprehensively after DAI was modified by naphthalene sulphonate esterification. This indicates that this kind of derivatives may have relatively good absorption and transport characteristics and biological activities *in vivo*. The research on the biological activities of the new derivatives (DD4 and DD5) are ongoing in our laboratory.

Data accessibility. The raw experimental data is provided as electronic supplementary material.

Authors' contributions. Y.J. performed the data analyses and wrote the manuscript; J.P. performed the experiment; X.Y. and H.H. contributed to analysis and manuscript preparation; L.G., J.Y. helped perform the analysis with constructive discussions; Y.P. contributed to the conception of the study.

Competing interests. We declare we have no competing interests.

Funding. This work was supported by a grant from the National Natural Sciences Foundation of China (grant no. 81660541) and the Natural Science Foundation of Jiangxi Province, China (grant no. 20202BABL203017).

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
