## [Reviewer comments · Royal Society Open Science]

Review History

RSOS-201475.R0 (Original submission)

Review form: Reviewer 1

Is the manuscript scientifically sound in its present form?

No

Are the interpretations and conclusions justified by the results?

No

Is the language acceptable?

No

Do you have any ethical concerns with this paper?

No

Have you any concerns about statistical analyses in this paper?

No

Recommendation?

Major revision is needed (please make suggestions in comments)

Comments to the Author(s)

How does solubility in ethylacetate relate to enhanced biological properties? The claim of the novelty of the acylation and alkylation products of DAI is unwarranted.

Why have the authors synthesized napsylates? What is the pharmacological significance or rationale for such a synthetic proposition?

The authors have mentioned, "their pharmaceutical properties could be increased through changes in the structure in order to enhance its biological activity." But where is the backing of biological assays to establish that? provide more convincing biological with DAI as a reference.

Further, the authors have claimed, "In vitro metabolic studies can reasonably predict the in vivo pharmacokinetic behaviour of candidate compounds by using the in vitro metabolic parameters of candidate compounds in the early stage of new drug development." On the contrary, in vitro inferences are often reversed in the in vivo conditions. Such a claim is extremely unprofessional. Please provide more convincing biological assays wherein both in vitro results tally with in vivo conditions.

Scheme 3 is confusing as a "possible metabolic pathway" ... Is it based on assumption? Clarify.

In the conclusion the authors claim the novelty of the derivatives based on the following observation:

"It indicates that this kind of derivatives may have relatively good absorption and transport characteristics and biological activities in vivo."

There's no report of in vivo experiments in the paper.

Also, the words like in vitro and in vivo should be in italics.

The authors have claimed pharmacological evaluation through HPLC but failed to convince how HPLC can establish such properties. Justify the claim.

The grammatical mistakes and the use of lackluster English are repetitive, making the manuscript weak in many instances. For example, "The cultures were cultured terminated at different points of 0.5, 1, 6, 12, 24 and 48 h respectively." How does one culture the cultures? There are many such instances. Revise the language thoroughly.

The format of references is inconsistent in ref nos 26, 27 etc.

Review form: Reviewer 2 (Shuihong Li)

Is the manuscript scientifically sound in its present form?

Yes

Are the interpretations and conclusions justified by the results?

Yes

Is the language acceptable?

No

Do you have any ethical concerns with this paper?

No

Have you any concerns about statistical analyses in this paper?

No

Recommendation?

Major revision is needed (please make suggestions in comments)

Comments to the Author(s)

The research addresses the enhanced solubility and cell uptake properties of the synthesized DAI derivatives by lipophilic modification, DD4 and DD5. The manuscript requires a major revision before publication in Roy Soc Open Sci.

A few comments:

- 1) The overall quality of the text is poor, the manuscript needs a native English speaker revision.
- 2) Please list the full name of "HAVSMCs" at the first time it appears.
- 3) Please number the carbon atoms of the compounds.
- 4) Line 19-36: The abstract can reflect all aspects of the work, but it is a bit long and too detailed. The authors should rephrase the abstract for a more concise expression.
- 5) As far as I'm aware, daidzein as a biologically active natural product has been used for curing angiocardopathy. But whether the synthesized daidzein napsylates have significant toxic side effects remain unknown. So, in vitro cytotoxicity assays of DAI, DD4 and DD5 on normal human cells (e.g. MRC-5/9, IMR-90, A549, HeLa cells) need be performed.
- 6) Please attach the 1H-NMR spectrum of DD4 and DD5 in supporting information; The coupling constant should be retained to a decimal place; In line 34, the peak splitting of 4'-OCH₂- and -CH₃ should be "q" and "t" instead of "m", respectively; 2-H, 8-H should be "s", please reassign the 2-H, 8-H of DD4 and DD5.
- 7) Is there no previous literature about HPLC detection methods for detecting DAI or its derivative? If any, the results of methodology verification (section 4.5) will be no need to write it out.
- 8) Conclusion may look a bit verbose, pls reword it.
- 9) There are some minor grammatical errors that need to be corrected. E.g. some words of page 1: Page 1, Line 23-24: Please rephrase: "and their possible metabolic pathways were inferred in vitro". Metabolic pathways should refer to "in vivo".
Line 29: Should "mono glucuronide" be changed into "mono glucuronate" ?
Page 1, Line 36: "The research on biological activities of the new derivatives (DD4, DD5) are undergoing in our laboratory". Here "are" should be changed into "is".
Page 1, Line 48: Should "hydrophilityis" be changed to "hydrophily is" ?
Page 1, Line 51: Should "in rats" be changed into "to rats" ?
Page 1, Line 51: "monohydroxylated"; Page 1, Line 59: "derivativesby"; Page 2, Line 20: "acidrespectively".....In these places, there should be a space between two words. Of course, there are many similar errors around the text.

Decision letter (RSOS-201475.R0)

Dear Dr Peng:

Title: Study on Pharmacological Properties and Cell Absorption Metabolism of Novel Daidzein Napsylates
Manuscript ID: RSOS-201475

The editor assigned to your manuscript has now received comments from reviewers. We would like you to revise your paper in accordance with the referee and Subject Editor suggestions which can be found below (not including confidential reports to the Editor). Please note this decision does not guarantee eventual acceptance.

Please submit your revised paper before 15-Oct-2020. Please note that the revision deadline will expire at 00.00am on this date. If we do not hear from you within this time then it will be assumed that the paper has been withdrawn. In exceptional circumstances, extensions may be possible if agreed with the Editorial Office in advance. We do not allow multiple rounds of revision so we urge you to make every effort to fully address all of the comments at this stage. If deemed necessary by the Editors, your manuscript will be sent back to one or more of the original reviewers for assessment. If the original reviewers are not available we may invite new reviewers.

RSC Associate Editor:
Comments to the Author:
(There are no comments.)

RSC Subject Editor:

Comments to the Author:
(There are no comments.)

Reviewers' Comments to Author:
Reviewer: 1

Comments to the Author(s)

How does solubility in ethylacetate relate to enhanced biological properties? The claim of the novelty of the acylation and alkylation products of DAI is unwarranted.

Why have the authors synthesized napsylates? What is the pharmacological significance or rationale for such a synthetic proposition?

The authors have mentioned, "their pharmaceutical properties could be increased through changes in the structure in order to enhance its biological activity." But where is the backing of biological assays to establish that? provide more convincing biological with DAI as a reference.

Further, the authors have claimed, "In vitro metabolic studies can reasonably predict the in vivo pharmacokinetic behaviour of candidate compounds by using the in vitro metabolic parameters of candidate compounds in the early stage of new drug development." On the contrary, in vitro inferences are often reversed in the in vivo conditions. Such a claim is extremely unprofessional. Please provide more convincing biological assays wherein both in vitro results tally with in vivo conditions.

Scheme 3 is confusing as a "possible metabolic pathway" ... Is it based on assumption? Clarify.

In the conclusion the authors claim the novelty of the derivatives based on the following observation:

"It indicates that this kind of derivatives may have relatively good absorption and transport characteristics and biological activities in vivo."

There's no report of in vivo experiments in the paper.

Also, the words like in vitro and in vivo should be in italics.

The authors have claimed pharmacological evaluation through HPLC but failed to convince how HPLC can establish such properties. Justify the claim.

The grammatical mistakes and the use of lackluster English are repetitive, making the manuscript weak in many instances. For example, "The cultures were cultured terminated at different points of 0.5, 1, 6, 12, 24 and 48 h respectively." How does one culture the cultures? There are many such instances. Revise the language thoroughly.

The format of references is inconsistent in ref nos 26, 27 etc.

Reviewer: 2

Comments to the Author(s)

The research addresses the enhanced solubility and cell uptake properties of the synthesized DAI derivatives by lipophilic modification, DD4 and DD5. The manuscript requires a major revision before publication in Roy Soc Open Sci.

A few comments:

- 1) The overall quality of the text is poor, the manuscript needs a native English speaker revision.
- 2) Please list the full name of "HAVSMCs" at the first time it appears.

- 3) Please number the carbon atoms of the compounds.
- 4) Line 19-36: The abstract can reflect all aspects of the work, but it is a bit long and too detailed. The authors should rephrase the abstract for a more concise expression.
- 5) As far as I'm aware, daidzein as a biologically active natural product has been used for curing angiocardopathy. But whether the synthesized daidzein napsylates have significant toxic side effects remain unknown. So, in vitro cytotoxicity assays of DAI, DD4 and DD5 on normal human cells (e.g. MRC-5/9, IMR-90, A549, HeLa cells) need be performed.
- 6) Please attach the ¹H-NMR spectrum of DD4 and DD5 in supporting information; The coupling constant should be retained to a decimal place; In line 34, the peak splitting of 4'-OCH₂- and -CH₃ should be "q" and "t" instead of "m", respectively; 2-H, 8-H should be "s", please reassign the 2-H, 8-H of DD4 and DD5.
- 7) Is there no previous literature about HPLC detection methods for detecting DAI or its derivative? If any, the results of methodology verification (section 4.5) will be no need to write it out.
- 8) Conclusion may look a bit verbose, pls reword it.
- 9) There are some minor grammatical errors that need to be corrected. E.g. some words of page 1: Page 1, Line 23-24: Please rephrase: "and their possible metabolic pathways were inferred in vitro". Metabolic pathways should refer to "in vivo".
Line 29: Should "mono glucuronide" be changed into "mono glucuronate" ?
Page 1, Line 36: "The research on biological activities of the new derivatives (DD4, DD5) are undergoing in our laboratory". Here "are" should be changed into "is".
Page 1, Line 48: Should "hydrophilityis" be changed to "hydrophily is" ?
Page 1, Line 51: Should "in rats" be changed into "to rats" ?
Page 1, Line 51: "monohydroxylated"; Page 1, Line 59: "derivativesby"; Page 2, Line 20: "acidrespectively"In these places, there should be a space between two words. Of course, there are many similar errors around the text.

Author's Response to Decision Letter for (RSOS-201475.R0)

See Appendix A.

RSOS-201475.R1 (Revision)

Review form: Reviewer 2 (Shuihong Li)

Is the manuscript scientifically sound in its present form?

Yes

Are the interpretations and conclusions justified by the results?

Yes

Is the language acceptable?

Yes

Do you have any ethical concerns with this paper?

No

Have you any concerns about statistical analyses in this paper?

No

Recommendation?

Accept as is

Comments to the Author(s)

I believe that you have seriously revised your manuscript according to the critical points. Overall, the revision is well written and there is no findable error in the entire paper.

Decision letter (RSOS-201475.R1)

Dear Dr Peng:

Title: Study on Pharmacological Properties and Cell Absorption Metabolism of Novel Daidzein Napsylates

Manuscript ID: RSOS-201475.R1

It is a pleasure to accept your manuscript in its current form for publication in Royal Society Open Science. The chemistry content of Royal Society Open Science is published in collaboration with the Royal Society of Chemistry.

RSC Associate Editor:
Comments to the Author:
(There are no comments.)

RSC Subject Editor:
Comments to the Author:
(There are no comments.)

Reviewer(s)' Comments to Author:
Reviewer: 2

Comments to the Author(s)

I believe that you have seriously revised your manuscript according to the critical points. Overall, the revision is well written and there is no findable error in the entire paper.

Appendix A

Dear Dr Smith,

We deeply appreciate you for your help in processing the review of our manuscript (Manuscript ID: RSOS-201475) in such tough time. We have carefully read the thoughtful comments from you and reviewers and found that those suggestions are helpful for us to improve our manuscript. On the basis of the enlightening questions and helpful advice, we have now completed the revision of our manuscript. We employed an English-language editing service, LetPub, to polish our wording. Certification is attached. The itemized responses to the reviewers' comments are listed in the succeeding sheets.

We hope that all these corrections and revisions would be satisfactory.

Thank you again and best regards!

Sincerely,

Dr. You Peng

Responses to comments of Reviewers

Thank you for your serious and constructive comments on our manuscript. According to your suggestions, the manuscript has been revised as a letter to editor, The revisions we have made are as follows.

Reviewer: 1

- How does solubility in ethylacetate relate to enhanced biological properties? The claim of the novelty of the acylation and alkylation products of DAI is unwarranted. Why have the authors synthesized napsylates? What is the pharmacological significance or rationale for such a synthetic proposition?

Reply: Thank you for your constructive and helpful suggestion.

Solubility of daidzein affects its bioavailability. The lipid solubility of drugs was increased to improve their bioavailability, which could enhance biological properties. We used the word “novel” because no others synthesised the compounds as far as we know. The aim that we synthesized these two compounds was to improve the pharmacokinetic properties of daidzein. We design target molecules according to the principle of pro-drug in pharmaceutical chemistry.

- The authors have mentioned, "their pharmaceutical properties could be increased through changes in the structure in order to enhance its biological activity." But where is the backing of biological assays to establish that? provide more convincing biological with DAI as a reference.

Reply: Thank you for your constructive and helpful suggestion.

The changes of structure could increase solubility to improve the bioavailability of the drug, which may enhance its biological activity. The more details could be found in the reference 4 (Y. Peng, Y. N. Shi, H. Zhang, Y. Mine and R. Tsao, Anti-inflammatory and anti-oxidative activities of daidzein and its sulfonic acid ester derivatives, *J.*

Funct. Foods, 2017, **35**, 635-640.).

- Further, the authors have claimed, "In vitro metabolic studies can reasonably predict the in vivo pharmacokinetic behaviour of candidate compounds by using the in vitro metabolic parameters of candidate compounds in the early stage of new drug development." On the contrary, in vitro inferences are often reversed in the in vivo conditions. Such a claim is extremely unprofessional. Please provide more convincing biological assays wherein both in vitro results tally with in vivo conditions.

Reply: Thank you for your constructive and helpful suggestion.

We apologize for the mistake. The sentence was cited in the reference 24 and we have added it in the revision.

- Scheme 3 is confusing as a "possible metabolic pathway" ... Is it based on assumption? Clarify.

Reply: Thank you for your constructive and helpful suggestion.

The metabolic pathway was deduced on the basis of the experimental results, such as the metabolic components of the system, general principle of organic biochemical reaction in cell (like enzymatic hydrolysis) *etc.*

- In the conclusion the authors claim the novelty of the derivatives based on the following observation: "It indicates that this kind of derivatives may have relatively good absorption and transport characteristics and biological activities in vivo." There's no report of in vivo experiments in the paper. Also, the words like in vitro and in vivo should be in italics.

Reply: Thank you for your constructive and helpful suggestion.

It was inferred that the derivatives should have good biological activities according to **the principle of pharmacokinetics**. The biological activities *in vivo* experiments are still in process. The fonts of *in vitro* and *in vivo* have been changed to italics in the revision.

- The authors have claimed pharmacological evaluation through HPLC but failed to convince how HPLC can establish such properties. Justify the claim.

Reply: Thank you for your constructive and helpful suggestion.

The solubilities and $\lg P$ of DD4 and DD5 were determined by HPLC, and their molecular polar surface area and polarisability, etc., were calculated by the software ChemAxon 16.1.18. The details was seen in Section 3.

- The grammatical mistakes and the use of lackluster English are repetitive, making the manuscript weak in many instances. For example, "The cultures were cultured terminated at different points of 0.5, 1, 6, 12, 24 and 48 h respectively." How does one culture the cultures? There are many such instances. Revise the language thoroughly.

Reply: Thank you for your constructive and helpful suggestion.

We have rewritten the sentence what you mentioned, revised the language and employed an English-language editing service, LetPub, to polish our wording. Certification is submitted.

- The format of references is inconsistent in ref nos 26, 27 etc.

Reply: Thank you for your constructive and helpful suggestion.

We have changed "*Frontiers in Pharmacology*" to "*Front.Pharmacol.*" in reference 27 in the revision. References of 26 and 27 are from open access, so they only have one page number.

Reviewer: 2

- 1) The overall quality of the text is poor, the manuscript needs a native English speaker revision.

Reply: Thank you for your constructive and helpful suggestion.

We employed an English-language editing service, LetPub, to polish our wording. Certification is submitted.

- 2) Please list the full name of “HAVSMCs” at the first time it appears.

Reply: Thank you for your constructive and helpful suggestion.

We have listed the full name of “HAVSMCs” as its first appearance in the abstract of the revision.

- 3) Please number the carbon atoms of the compounds.

Reply: Thank you for your constructive and helpful suggestion.

The carbon atoms of DAI, DD4 and DD5 have been numbered in the revision.

- 4) Line 19-36: The abstract can reflect all aspects of the work, but it is a bit long and too detailed. The authors should rephrase the abstract for a more concise expression.

Reply: Thank you for your constructive and helpful suggestion.

We have rewritten the section of abstract and tried to make it as concise as possible.

- 5) As far as I'm aware, daidzein as a biologically active natural product has been used for curing angiocardopathy. But whether the synthesized daidzeinnapsylates have significant toxic side effects remain unknown. So, in vitro cytotoxicity assays of DAI, DD4 and DD5 on normal human cells (e.g. MRC-5/9, IMR-90, A549, HeLa cells) need be performed.

Reply: Thank you for your constructive and helpful suggestion.

The cytotoxicity of DD4 and DD5 was determined by MTT assay in our lab before the drug cell absorption experiment, and it was found that the cell inhibition rate of the two compounds was less than 10% at the concentration of 300 ug/mL. It is considered that the compound is not toxic to vascular smooth muscle cells at this concentration. Therefore, the dosing concentrations of DD4 and DD5 were 171 and 120 ug/mL, respectively, to investigate the absorption and metabolism of drugs by cells.

- 6) Please attach the ¹H-NMR spectrum of DD4 and DD5 in supporting information;

The coupling constant should be retained to a decimal place; In line 34, the peak splitting of 4'-OCH₂- and -CH₃ should be “q” and “t” instead of “m”, respectively; 2-H, 8-H should be “s”, please reassign the 2-H, 8-H of DD4 and DD5.

Reply: Thank you for your constructive and helpful suggestion.

We apologize for the mistakes in the data of ¹H-NMR spectrum of DD4 and DD5. The ¹H-NMR spectrum of DD4 and DD5 has been submitted as supplementary materials as you required. We have reassigned the relevant H of DD4 and DD5 as you required in the revision.

- 7) Is there no previous literature about HPLC detection methods for detecting DAI or its derivative? If any, the results of methodology verification (section 4.5) will be no need to write it out.

Reply: Thank you for your constructive and helpful suggestion.

The section 4.5 was summarized based on the experimental procedure in the reference (Wang Shaoyu, Du Zongliang, Li Ruixia, Wu Dacheng, Tai Yuanlin. Experimental Measurement and Correlation of Solubility of Quercetin in Absolute Ethanol, The Global Seabuckthorn Research and Development, 2004, 2(3): 12-15.). However, we chose to retain it in the revision because it is an integral part of our experimental method. We have added it as reference 15 in the revision.

- 8) Conclusion may look a bit verbose, pls reword it.

Reply: Thank you for your constructive and helpful suggestion.

We have rewritten the conclusion and tried to make it as concise as possible.

- 9) There are some minor grammatical errors that need to be corrected. E.g. some words of page 1:

Page 1, Line 23-24: Please rephrase: “and their possible metabolic pathways were inferred in vitro”. Metabolic pathways should refer to “in vivo”.

Line 29: Should “mono glucuronide” be changed into “mono glucuronate”?

Page 1, Line 36: “The research on biological activities of the new derivatives (DD4, DD5) are undergoing in our laboratory”. Here “are” should be changed into “is”.

Page 1, Line 48: Should “hydrophilityis” be changed to “hydrophily is” ?

Page 1, Line 51: Should “in rats” be changed into “to rats” ?

Page 1, Line 51: “monohydroxylated”; Page 1, Line 59: “derivativesby”; Page 2, Line 20: “acidrespectively”.....In these places, there should be a space between two words. Of course, there are many similar errors around the text.

Reply: Thank you for your constructive and helpful suggestion.

All grammatical errors have been corrected including what you pointed out and other mistakes, we apologize for our errors.

We would like to thank the referees again for taking the time to review our manuscript.